# Interval from Oestrus to Ovulation in Dairy Cows—A Key Factor for Insemination Time: A Review

**DOI:** 10.3390/vetsci11040152

**Published:** 2024-03-29

**Authors:** Fabio De Rensis, Eleonora Dall’Olio, Giovanni Maria Gnemmi, Padet Tummaruk, Melania Andrani, Roberta Saleri

**Affiliations:** 1Department of Veterinary—Medical Science, University of Parma, Via del Taglio 10, 43126 Parma, Italy; fabio.derensis@unipr.it (F.D.R.); roberta.saleri@unipr.it (R.S.); 2Bovinevet Internacional SL. Bovine Reproduction Ultrasonography & Herd Management Huesca (ES), 22006 Huesca, Spain; giovanni.gnemmi@bovinevet.com; 3Department of Obstetrics, Gynecology and Reproduction, Faculty of Veterinary Science, Centre of Excellence in Swine Reproduction, Chulalongkorn University, Bangkok 10310, Thailand; padet.t@chula.ac.th

**Keywords:** dairy cattle, oestrus, ovulation, AI

## Abstract

**Simple Summary:**

This review delves into the oestrus-to-ovulation interval, the potential for ovulation time prediction, and the optimal timing for insemination in dairy cows. Typically, oestrus lasts 10–20 h in dairy cows, with variations attributed to oestrus detection methods and observation frequency. Most cows ovulate around 24–33 h after oestrus onset and 15–22 h after the end of oestrus. The interval from the preovulatory luteinising hormone (LH) surge to ovulation range is approximately 21–30 h, with ovulation occurring at a follicle diameter averaging 18–20 mm. Correctly identifying oestrus onset allows for efficient artificial insemination using the “a.m.–p.m. rule”. However, excessively long or short oestrus-to-ovulation intervals may compromise fertility. Heat stress during warm seasons can increase the variability of the oestrus-to-ovulation interval. In cases where insemination timing risks being too early or late relative to ovulation, GnRH administration may be considered to optimize fertility outcomes.

**Abstract:**

This review describes the oestrus-to-ovulation interval, the possibility of predicting the time of ovulation, and the optimum time for insemination relative to oestrus in dairy cows. The duration of oestrus in dairy cows is approximately 8–20 h, with differences possibly related to the methods of oestrus detection and the frequency of observations. Most cows ovulate approximately 24–33 h after the onset of oestrus and 15–22 h after the end of oestrus. The interval from the preovulatory luteinising hormone (LH) surge to ovulation is approximately 4–30 h. Ovulation occurs when follicle diameter averages 18–20 mm. When it is possible to correctly determine the beginning of oestrus, artificial insemination can be performed utilizing the “a.m.–p.m. rule”, and only one insemination may be applied. In cows with too long or too short oestrus-to-ovulation intervals, fertility can be compromised. One important factor that can alter the oestrus-to-ovulation interval is acute or chronic heat stress during the warm season. When there is a risk that insemination may occur too early or too late with respect to the time of ovulation, GnRH administration can be considered.

## 1. Introduction

In dairy cows, successful insemination must occur at the correct time relative to oestrus. Determining the exact moment of ovulation is challenging due to variations in oestrus detection strategies and the timing of insemination relative to ovulation. There are several consequences if insemination is performed to early or too late after ovulation. In fact, when insemination occurs too early, the sperm ages, and by the time ovulation takes place it may not be able to fertilize the ovum [1]. Conversely, when insemination takes place too late, the egg may age, compromising fertilization and the formation of a viable embryo [2]. Therefore, it is essential to identify when ovulation occurs. Inefficient timing of artificial insemination (AI) results in lost lifetime and milk yield, a decreased number of calves born per lifetime, excessive days open, and increased reproductive culling.

Several factors can affect fertility when AI is utilized at spontaneously occurring oestrus, like the intensity of oestrous expression [3,4] and behavioural changes associated with oestrus, such as walking activity [3,5], neck movement [3,6,7], and lying behaviour [8]. Data from the literature indicate that cows with reduced oestrous expression may have compromised fertility due to improper ovulation times or ovulation failure [7,9]. Recently, it has come to light that using GnRH during AI can enhance the fertility of cows with reduced oestrous expression [10]. Therefore, this review describes the oestrus-to-ovulation interval (OE–OV interval) and the possibility of predicting the time of ovulation to make it possible to inseminate at the correct time. Some factors than can affect the OE–OV interval like seasonal heat stress are also discussed. Finally, it is determined that GnRH administration can reduce the variability of the oestrus-to-ovulation interval.

## 2. Duration of Oestrus and the Interval from Oestrus to Ovulation (OE–OV Interval)

The best approach to determine the optimum time for AI is to know the time of ovulation. However, to date, there are no direct signs that can accurately predict when a preovulatory follicle is going to ovulate. Nevertheless, there are some indirect signs that can help to predict the time of ovulation, most of which are related to oestrus behaviour. Oestrus behaviour in dairy cows has been reviewed [11]. Briefly, there are primary and secondary signs of oestrus behaviour. The primary sign is the standing reflex, during which the cow makes no attempt to escape when mounted [12,13]. This is an important sign to predict the time of ovulation, but unfortunately, standing oestrus is not displayed during all oestrus cycles. In some studies, it has been detected only in 50–58% of oestrus periods [14,15,16], sometimes even at a lesser degree (20%), especially if there is only one cow in the herd that is in oestrus [17]. The secondary signs are numerous (for a review, see [18]), and they generally increase significantly 1 to 3 h before the start of standing oestrus [18,19], but with high variability among animals, which reduces their efficacy as predictors of ovulation time.

Due to the importance of oestrus detection, several monitoring systems have been developed. The observation of oestrus behaviour and activity monitoring are the most commonly utilized practices. Recently, other measures or traits have been investigated, such as the evaluation of rumination activity, eating activity, milk output, and body temperature.

In addition, significant research progress has been made in monitoring cows with electronics, biosensors, and computers [20]. All these systems have some efficacy, but standing oestrus remains the best predictor for the time of ovulation [21].

A direct comparison among studies on the duration of oestrus remain difficult, as studies differ in the type of oestrus detection device utilized and their experimental design (e.g., the inclusion of timed AI), but, generally, the average duration of oestrus is approximately 14 h with a range of 8 to 20 h [17,22,23,24,25,26]. For example, mean intervals of 11.8 ± 4.4 h (measured by visual observation) [18], 16.1 ± 4.7 (measured by an accelerometer) [24], 7.1 ± 5.4 h (measured by radiotelemetry) [23], and 14.94 ± 0.33 (measured by automated activity monitoring, AAM) have been reported [25,27,28,29].

Differences in age, herd size, management practices, the frequency of observations, and the definition of oestrus onset may explain most of the variations in oestrus duration. For example, regarding the parity of cows, some studies have reported that the duration of oestrus tends to be longer for primiparous (13.6 ± 4.8 h) compared to multiparous animals (10.8 ± 4.4 h), while in other studies it tends to be longer for multiparous (13.6 ± 2.0 h) compared to primiparous (7.4 ± 1.0 h) cows, or no differences have been detected (for a review, see [18]). Other factors affecting oestrous expression in lactating dairy cows have been identified, such as flooring [29], parity and body condition score (BCS) [24,25], and milk yield [30].

The duration and intensity of oestrus expression have been found to be strongly associated with the risk of pregnancy and embryonic development in lactating dairy cows [3,6,31].

Cows with an oestrus event of high intensity or long duration achieved greater pregnancy per AI (P/AI) compared with cows with an oestrus event of low intensity or short duration [3,4,25,32].

Another factor that can lead to a shorter and less intense expression of oestrus in high-yielding dairy cows is the elevated metabolism of steroid hormones that occurs in these animals due to their high milk production [5,30,33]. Interestingly, lower oestrus intensity [7] and a shorter duration of oestrus [9,24] have been associated with shorter OE–OV intervals and increased ovulation failure [4,7,8].

Additionally, some animals may ovulate without exhibiting standing oestrus (silent ovulation), or may show oestrus without ovulating [34,35]. Some studies have reported that the incidence of silent ovulation at first ovulation after calving ranges from 50% to 94% [14,36], and at second ovulation from 6% to 50% [34,35]. In cows under heat stress during the warm period, the percentage of cows that failed to ovulate by day 50 post-partum was 29%, while during the cold period it was 10% [37,38].

### Interval from Oestrus to Ovulation (OE–OV Interval)

Direct comparison among studies determining the OE–OV interval remains challenging due to variations in the definition of oestrus onset and the experimental designs used to determine ovulation time. However, generally, most dairy cows ovulate approximately 24 and 33 h after oestrus onset [9,16,17,18,39,40,41,42,43] (Table 1) with no differences detected between primiparous (28.7 ± 3.9 h) and multiparous (27.0 ± 4.1) cows [44,45]. In a detailed study [43], it was observed that in a population of 106 cows, half of them (47%) exhibited a normal OE–OV interval (26 to 30 h), 25% exhibited a short OE–OV interval (22 to 25 h), 17% showed a long OE–OV interval (31 to 35 h), and 10% showed a very long OE–OV interval (>36 h). Some studies have described a relatively long OE–OV interval of 38.5 ± 3.05 [46], with the authors suggesting that these differences in the duration of the OE–OV interval could be related to factors associated with follicular growth.

These differences in the length of the OE–OV interval also depend on the method used to detect the onset of oestrus. For example, the onset of oestrus can be determined by the following methods:Visual detection of the standing reflex: the average OE–OV interval is between 27 and 30 h [14,40,41]. In tie-stall-housed cows, Suthar et al. [47] reported a shorter OE–OV interval (20 h).Automatic activity monitors like pedometers: the average OE–OV interval is about 27 to 33 h [48,49]. However, with the use of collar-mounted systems (accelerometers), it is about 21 to 35 h [9,24,25,39].Pressure-sensing systems: the OE–OV interval is about 21–32 h, with most animals detected in oestrus 26.4 h [9] or 27.6 h [40] before ovulation.

Among the factors that can modify the OE–OV interval, it is worth mentioning the inflammatory process due to uterine and mammary infections that can interfere directly or indirectly with ovarian function and the OE–OV interval. In cows subjected to intrafollicular treatment with lipopolysaccharides, there is a longer OE–OV interval compared to untreated cows [50,51]. Previous observations are confirmed by these data that lipopolysaccharides released into the circulation during infectious diseases can impact the timing of ovulation in dairy cow [52,53,54].

When considering the end of oestrus instead of the onset of oestrus, earlier studies reported that ovulation generally occurs 9 to 12 h after the end of oestrus [55,56], while more recent reports indicate that the end of the oestrus-to-ovulation interval range is between 15 and 22 h [9,17,39] (Table 1). Indeed, the end of standing oestrus appears to be a more accurate predictor of ovulation time than the onset of standing oestrus [26,57].

## 3. Relationship between Preovulatory LH Surge and Ovulation (LH–OV Interval)

The preovulatory LH surge is a key factor to determine the timing of ovulation. The synthesis and secretion of oestradiol by the preovulatory follicle during the last 3 to 4 days of the oestrus cycle stimulate the pulsatile release of GnRH from the hypothalamus into the hypophyseal portal system, leading to the subsequent release of LH from the anterior pituitary into systemic circulation. Elevated circulating concentrations of LH induce a cascade of events within the mature follicle, culminating in follicular rupture and ovulation [36,58]. Recent studies also provide direct evidence of the existence of an ovulation-inducing factor (OIF) present in seminal plasma. This factor acts systemically rather than locally and is a potent stimulant of LH secretion and ovulation [58].

The interval between the onset of oestrus and the preovulatory LH surge is approximately 3–5 h [17], with a mean of 2.8 ± 0.4 h [43]. The interval between the preovulatory LH surge and ovulation is 21–30 h [59,60], with some studies reporting a mean of 25.3 ± 0.6 h [17], 29 ± 1.5 h [46], 28.5 ± 1.4 h [28], and 25.5 ± 0.5 h [43]. The differences between the length of the interval LH surge and ovulation could be due to variations in follicular development, feed intake, energy balance, days in milk, parity, and season [43,46].

Even if an extended OE–OV interval can be associated with alterations in the plasma concentrations of progesterone and oestradiol [43], most of the variation can be attributed to variations in LH surge occurrence [46], and therefore treatment with GnRH at the onset of oestrus may reduce this variability [43,61].

## 4. The Timing of AI

### 4.1. Timing of AI Relative to Ovulation

Inseminations performed at the wrong time reduce pregnancy rates [48,62,63]. Therefore, the timing of insemination relative to ovulation is critical for fertility. Successful fertilisation requires the presence of spermatozoa in the uterine tubes at the correct time to fertilize the ovum [20,22,23,24,64]. Thus, for a correct timing of insemination, the lifespan of bull spermatozoa must be considered, which ranges between 24 and 30 h [65], as well as the time required for sperm transport to the site of fertilization and their capacitation, which averages 6–8 h [1,64,65,66]. Another factor to consider is the lifespan of the ovulated ovum that in cow ranges 8–12 h, and the optimum period for the fertilisation of the ovum, which is between 6 and 12 h after ovulation [36,67,68]. Thus, there is a window of at least 12 h for fertilisation [69]. Some studies reported an even wider window of fertility, suggesting that conception can occur at 40 h after the onset of oestrus and sperm can survive awaiting ovulation for 20–30 h or longer [70].

Because the fertilisation capacity of the oocyte decreases significantly 8 to 12 h post-ovulation [71], when AI takes place too late after the onset of oestrus, the oocyte remains in the oviduct for too long before fertilization, resulting in poor embryo quality [1,71]. In contrast, when AI occurs too early, more than 24 h before ovulation, many spermatozoa potentially die before the oocyte arrives at the ampulla, resulting in low fertilization rates [1,66,72].

Insemination should take place between 7 and 18 h [26,39,55,65,71] before ovulation, but successful inseminations have also been observed between 0 and 16 h [63] before ovulation.

Roleofs [71] reported that fertilization rates were significantly higher when AI was conducted between 12 and 24 h before ovulation (68%), compared to 0–12 (41%) or 24–36 h before ovulation (41%). A recent study reported [73] higher conception rates (63.0%) obtained by AI completed 6–18 h before ovulation, while for AI completed 30 h after ovulation or 6 h before ovulation, the conception rates decreased to 30.0% and 26.9%, respectively.

Considering the effect of insemination time on embryo quality, previous research did not find any differences when inseminations were performed 0–24 h before ovulation [72], with the highest percentage of good-quality embryos (89%) being recovered when AI was performed between 12 and 24 h before ovulation. However, AI performed after ovulation did not result in good embryos at all [26]. These data suggest that for embryo quality, better results are obtained when insemination occurs closest to ovulation, compared to the other times, for the best fertilization rate.

Thus, the highest pregnancy rates take place when inseminations occur before ovulation. However, in dairy cows, several factors can influence oestrus detection, such as the season of insemination, heat stress, feed intake, energy balance, days in milk at AI, milk yield, and parity [11,12]. Consequently, applying AI at the proper time relative to ovulation can be difficult. López–Gatius [68] observed that 25% of inseminations are performed too early and 3.5% too late, with the negative effect of this asynchrony on pregnancy rate exacerbating during the warm period of the year in cows under heat stress.

How can we accurately predict the time of ovulation? As previously described, one of the best approaches is the detection of the onset of oestrus by visual observation of the standing reflex and by the continuous monitoring of cow activity. There are other methods with potential to predict the time of ovulation, such as determining vaginal electrical resistance [63,74] or infrared thermography [75], but at the moment, they cannot be used in practical herd management because of their specificities and the fact that their positive predictive values are lower compared with those obtained via the visual assessment of standing oestrus or automated activity monitors.

### 4.2. AI Timing Relative to Behavioural Oestrus

As previously described, behavioural signs can be used to predict the time of ovulation to determine the time for insemination. Thus, when considering the timing of AI related to behavioural oestrus, pioneering early works from the 1940s [55,68], as well as more recent work [16,24,28,49,58,62,63,72,74,76,77], indicate that insemination should take place 4–18 h after the first standing event, or during the first 6–8 h after the end of the last standing event [26,55,57,65]. These studies confirm that insemination can be performed utilizing the well-established “a.m.–p.m. insemination rule”, which recommends that cows observed in oestrus during the morning should be inseminated during the afternoon, and cows in oestrus during the afternoon should be inseminated the following morning. However, one study detected no differences among cows receiving a single AI during early or mid-oestrus compared to the “a.m.–p.m. insemination rule” [78].

However, even if the behavioural signs of oestrus are still the best predictor of the time of ovulation and a guide to the optimal time of insemination, several factors can affect optimal AI timing, like cow’s BCS [79], ovarian reserve [80], uterine size [81], and differences in milk yield [9,82]. Concerning cow parity, some reports indicate that the optimal interval from the onset of oestrus to AI is longer for primiparous cows compared to multiparous cows [9], while other reports [46,82] indicate the opposite. The precise causes of these discrepancies are unclear, but differences could be related to the pattern of follicular growth [46] or the oocyte quality. It has been reported that oocyte quality is superior in multiparous cows [83], and this could potentially allow a longer oestrus-to-ovulation interval. Another factor could be the larger uterus size in multiparous cows [81]. If AI is performed too early in multiparous cows with a large uterus, a greater proportion of sperm may die during passage through the uterus, resulting in fewer spermatozoa remaining active before the oocyte reaches the ampulla, compared to cows with a small uterus.

Considering the frequency of insemination, researchers have postulated that conception rate may be improved when the frequency of inseminations per day is increased from one to two because there is an increase in the proportion of inseminations performed in the peri-ovulatory period [63]. This could be particularly important in high-producing cows [7,30], in late-ovulating cows [84], in repeat breeding [85], in lame animals [86], in cows with low BCSs [87], and cows subjected to thermal stress [88,89].

However, when oestrus detection is accurate, the insemination technique is good, and semen fertility is high, insemination performed once a day does not show differences in pregnancy rate with insemination performed twice a day [76,90,91,92,93].

One of the challenges when considering optimal AI timing is the large variation in the interval of oestrus duration, as previously described. Thus, the end of oestrus could be the best predictor for determining ovulation time and the time for insemination. Furukawa et al. [26] reported a conception rate of 57.1% when AI was performed between 4 h before and 4 h after the end of oestrus, 37.7% when AI was performed 4–12 h before, and 30% when it was performed 12–20 h after the end of oestrus [26]. These data indicate that the best time for insemination would be between 4 and 12 h after the end of oestrus. This information is important especially when sex-sorted semen is utilized because sperm sexing (flow cytometry) can damage the cell itself, decreasing conception rates compared to unsexed semen [94]. For this reason, when sex-sorted semen is used, the insemination must be performed closer to ovulation compared to when unsexed semen is utilized.

## 5. Can Ovulation Be Predicted by Ultrasonography?

Transrectal examination of the bovine reproductive tract either by hand [95] or by ultrasonography [17] might allow for the detection of ovulation. As ovulation approaches, at rectal palpation, the follicle feels soft and separate from the remainder of the ovary [68,95,96].

Ultrasonography (such as echo–Doppler) evaluation in the management of reproduction in dairy cows is becoming fundamental and is used for the diagnosis of pregnancy, but also to evaluate follicular dynamics, the presence and development of the corpus luteum, the pathophysiology of the ovary and uterus, foetal sexing, the diagnosis of twins, embryonic and foetal death, and foetal malformations. Echo–Doppler also facilitates an improved selection of embryo and oocyte donor cows/heifers and recipients for embryo transfer. Ultrasonography might also be used to determine the time of ovulation. Typically, ovulation in cows occurs when the follicle diameter is approximately 18 to 25 mm (19.8 ± 6.1 mm) [57], with a maximum pregnancy rate observed in cows that ovulated from follicles around 14 mm in diameter (13.9 + 0.2 mm) [97]. This threshold was reported in beef cows [98] and *Bos indicus* [99]. Perry et al. [98] reported that the ovulation of follicles < 10.7 mm or >15.7 mm is less likely to support pregnancy compared to follicles with a size of 12.8 mm. The ultrasound evaluation of ovulation time in practice may have some limitations because it requires a frequency of examination of at least twice a day. The possibility that manipulation of the ovary might alter the timing of ovulation, especially when the technician lacks experience, must also be considered [67].

Colour Doppler ultrasonography has been used to investigate variations in local blood flow in individual ovulatory follicles. An increase in “blood flow velocity” coincided with the initiation of the LH surge, which remained unchanged until ovulation and then decreased 12 to 24 h after ovulation [100,101,102]. These data suggest that the determination of local blood flow within the follicular wall may be considered in the future to determine ovulation and, therefore, the best time for AI. However, currently, this approach is not practical.

## 6. Seasonal Heat Stress (HS) and the Oestrus-to-Ovulation Interval

Apart from factors such as energy balance, milk output, and the age of the cow, several environmental factors, such as temperature, humidity, and photoperiod, can play a role in modifying the OE–OV interval. In northern temperate areas during the summer season, chronic HS, as well as acute HS for a few days (heat waves), induce alterations in the neuroendocrine mechanisms that control follicular development and ovulation, leading to an increase in the variability of the OE–OV interval [103,104,105,106,107], which in turn increases the percentage of inseminations that are performed too early or too late [57]. Summer HS also decreases the expression of oestrus [106] and increases the likelihood of ovulation failure [103,108,109] by up to four times compared to the winter season [37].

Seasonal HS can alter the OE–OV interval due to the following factors:Altered plasma concentrations of LH [110], follicle-stimulating hormone (FSH) [97], oestradiol, and progesterone [108,109].Altered patterns of follicular waves: The average size of the first-wave dominant follicle is similar between heat-stressed and control cows, but it decreases in size more rapidly. Therefore, the second-wave dominant follicle emerges earlier, inducing alterations in follicular development that contribute to the ovulation of an aged follicle, thereby reducing fertility [111].Perturbations of the intrafollicular fluid milieu of the maturing oocyte and the periovulatory follicle [112,113,114].Alterations in the activity of numerous inflammatory-like factors that determine ovulation after LH surge. Ovulation is a complex mechanism that can be compared with an inflammatory response, and alterations in the characteristics of certain intrafollicular proteins and cytokines may alter the mechanism leading to ovulation [115,116]. These findings have been confirmed by an in vivo study where an increase in the intraovarian thermal conditions reduced the success of ovulation [117,118].A recent study in a tropical country demonstrated that oestradiol concentrations are lower in the hot, dry season (17.6 pg/dL) compared with the cold, rainy seasons (19.5 pg/dL), and this might modify the OE–OV interval [119].

Thus, HS can alter the mechanism of ovulation by affecting GnRH secretion, which alters LH and FSH secretion, follicular development, the pattern of the follicular wave, the characteristics of the intrafollicular milieu, and the release of inflammation factors associated with ovulation. All these events have a direct effect on the OE–OV interval, making it more difficult to apply AI at the best time for fertilisation.

Because it has been reported that the incidence of ovulation failure in cows under seasonal HS mainly depends on dysfunction in GnRH release or follicle development and not on the amount of LH required to induce ovulation [61,120], GnRH administration may be considered because it induces an LH preovulatory peak that determines ovulation and allows the time of insemination to be predicted more accurately [61,121,122].

## 7. Control of Ovulation by GnRH Administration

The administration of GnRH is directly linked to ovulation due to its effects on LH secretion and pulse frequency [61]. Kaim et al. [61] reported that GnRH administered within 3 h following the onset of oestrus increased the magnitude of LH surges. However, there are differences in the effects of GnRH on LH secretion depending on the time of GnRH administration with respect to the endogenous LH surge. When the GnRH administration coincides with the spontaneous LH peak, the resulting magnitude of the LH peak is more than twice as high as the spontaneous LH peak [61,123], but when GnRH is administered at AI, it does not increase the spontaneous LH surge but induces a second LH peak [124,125,126].

Morgan and Lean [125] conducted a meta-analysis on 40 experiments from 27 published papers and concluded that, although treatment with GnRH at insemination in most studies increased the conception rates, there is still a large variability, which could be due to the timing of GnRH administration with respect to ovulation and the age of the animals (heifers or cows). In two studies, GnRH administered early in oestrus had no effect on the conception rate [24,126].

### 7.1. Effect of GnRH Administration at Onset of Oestrus

GnRH administered at the onset of oestrus induces an LH peak [61] and decreases the time interval from OE to the LH surge peak (3.1 ± 0.5) compared with untreated animals (1.7 ± 0.4 h), but not the interval from the peak of LH surge to ovulation [61]. In this study, 100% of cows ovulated within 30 h of the onset of standing oestrus, whereas in untreated cows (controls), 24% had intervals higher than 30 h with a reduction in the number of successful inseminations. These data confirm previous studies in which GnRH administration at the onset of oestrus reduced the variation in the time interval to ovulation, indicating a possible preventive effect for ovulation failure [127,128].

GnRH treatment at oestrus significantly improved conception rate in cows under heat-stress conditions [122,129], in cows with a low BCS at AI, and in primiparous cows [61].

### 7.2. Effect of GnRH Administration at AI

Few studies have reported the effects of GnRH at AI on ovulation rates in spontaneous oestrus events. The administration of a GnRH analogue at the time of AI did not affect the interval from the onset of oestrus to ovulation [129], nor the proportion of cows that had ovulated by 7 d post-AI [10].

## 8. Effect of GnRH on Inducing Ovulation in a Fixed-Time Insemination Program

In the management of dairy cow reproduction, protocols for the synchronisation of ovulation that allow insemination to be carried out at a predicted time, without the need for oestrus detection (Ovsynch-based protocol), are utilized. These protocols are mainly based on a sequential administration of GnRH (to induce ovulation and a new follicular wave), prostaglandin (PGF)2α seven days later (to induce luteolysis), and GnRH again to induce ovulation. AI follows the second administration of GnRH [130,131,132,133].

In these protocols, the plasma concentration of oestradiol and progesterone at the moment of the first GnRH administration may affect the GnRH-induced LH secretion at the second GnRH administration, and therefore subsequent ovulation [61,134,135]. When the progesterone concentration exceeds 0.5 ng/mL or the oestradiol level is ≥4 pg/mL at the time of GnRH treatment, the LH plasma levels are reduced, and ovulation is suppressed [134]. Also, the presence and the size of the dominant follicle at the first GnRH administration are crucial for the induction of ovulation at the second GnRH administration.

Regarding the timing of insemination concerning the second GnRH administration in Ovsynch-based protocols, intervals of 8–24 h [131], 10–20 h [127], and 16–24 h [76,132,136] have been utilized.

In some circumstances, like in cows under seasonal HS, human chorionic gonadotropin (hCG) administration can be applied instead of GnRH because its LH-like effects [137,138,139] promote ovulation in cyclic and non-cyclic dairy cows [140]. In beef cows, it has been reported that 2500 IU of hCG increased the ovulatory response by 13.9 percentage points compared with 100 µg of GnRH [141].

## 9. Overall Conclusions

Correct AI timing in relation to ovulation is essential to optimise fertilisation. Table 2 provides an overview of the events related to the time of ovulation in dairy cows. Usually, ovulation occurs 27–30 h after the onset of oestrus and the optimal time for AI is 7–18 h before ovulation, i.e., 4–18 h after the onset of oestrus or 6–8 h after the end of oestrus. When the determination of oestrus is accurate, the traditional “a.m.–p.m. rule” with only one insemination is reliable for successful insemination. There are several factors that can help to predict ovulation time, such as a system that determines variations in cow activity, rumination, feed intake, rectal temperature, vaginal electrical resistance, progesterone levels, and infrared thermography. However, the most reliable approach is still the visual detection of the standing reflex. There have been some attempts to directly determine the exact time of ovulation by ultrasound examination, but to date, no reliable signs have been reported, apart from the average diameter of the preovulatory follicle, which should range between 18 and 20 mm at the time of ovulation. In some circumstances, such as cows under seasonal heat stress, GnRH can be used to induce ovulation and reduce the variability of the oestrus-to-ovulation interval.

## Figures and Tables

**Table 1 vetsci-11-00152-t001:** Example of the mean duration (± SD) for the onset of the oestrus-to-ovulation interval and the end of the oestrus-to-ovulation interval in dairy cows, as described in the literature.

Interval from Oestrus Onset to Ovulation	References
25.7 ± 0.4 h	[9]
26.4 ± 0.7 h	[9]
26.4 ± 1.5 h	[18]
27.6 ± 5.4 h	[40]
28.6 ± 6.0 h	[43]
28.7 ± 8.1 h	[24]
29.0 ± 0.6 h	[42]
30 ± 1.1 h	[28]
30.2 ± 0.6 h	[42]
30.6 ± 4.4 h	[18]
range 24–33 h	
**Interval from the End of Oestrus to Ovulation**	
15.3 ± 0.9 h	[9]
16.7 ± 1.1 h	[17]
17.3 ± 12.8 h	[39]
21.0 ± 0.9 h	[9]
Mean range 15–21 h	

**Table 2 vetsci-11-00152-t002:** Average duration of the oestrus-to-ovulation and oestrus–insemination interval and related events in dairy cows.

Hours from Onset of Oestrus	
Ovulation	Insemination
Onset of oestrus to ovulation: ~24–33 h End of oestrus to ovulation: ~15–22 h	Onset of oestrus to insemination: ~4–18 h End of oestrus to insemination: ~6–8 h
**Oestrus length: ~8–20 h**	
**Insemination should occur ~7–17 h before ovulation**	

## Data Availability

Data contained within this article.

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
