# Peer review of "Interval from Oestrus to Ovulation in Dairy Cows—A Key Factor for Insemination Time: A Review"

_vetsci, 2024, doi:10.3390/vetsci11040152_

Round 1

Reviewer 1 Report

Comments and Suggestions for Authors  

The proposed administration of GNRh is most justified, especially in the case of a prolonged interestrous period. It is definitely worth considering, especially for high-milk cows.

Author Response

Reviewer #1

  • The title reflects the content of the study. However, the title is no accurate and some chapters are unbalanced.

AU: We thank you for suggestion, the title has been modified and some sentences have been moved.

Reviewer 2 Report

Comments and Suggestions for Authors

The received ms vetsci-2876491 Interval from oestrus to ovulation in dairy cows, a key factor for fertilityby de Rensis et al. investigated more than 150 references to describe the OE-OV interval, the possibility to predict the ovulation and to adjust a sufficient AI. Special attention was given to the influence of heat stress and inflammatory processes within a cow and the application of sexed semen.. Needless to say that there is still a growing interest to understand reproductive functions and to organize an efficient reproductive management on the basis of physiological signals. The special asset of this study is to lighten the relationship between “invisible” hormonal mechanisms (LH-peak) distinguished animal behavior and reproduction management. Most of the literature is focused on dairy cows.

The review is well performed and written. The title reflects the content of the study. However, the title is no accurate and some chapters are unbalanced.

Major concerns:

The OE-OY interval is a irrefutable physiological fact – only its consideration is a key factor for fertility within our ”artificial driven”  reproduction management.

Figure 1 is a copy of a textbook diagram without critical reflection. A) The onset of oestrus does not start with the LH-peak. B) The spermatozoa and oocyte lifespans are depended from many different factors; these parameter are not the core of this review – therefore the reviewer respect the very schematic character of this figure. However, to rely on references which are more than 50-60 years old is questionable.

The authors raise the question regarding the conception rates following AI done twice or single a day without clear solution. A scientific review should improve real questionable practical situations. This chapter (L210-414) should be improved.

Why should be done an US-examination of the ovulation every 3 hours (L244) ? It is interesting to evaluate the ovulation – however this interval is questionable as a key factor for fertility (title of this review).

Minor concerns:

L 136 in combination with Fig. 1 is confusing.

The passages “sexed semen” (L220-225) and “inflammatory processes (L293-304) are unbalanced in comparison to the chapter “heat stress” and not identifiable in the subtitles. Should be extended or removed.

The application of hCG is an appropriate approach to stimulate the ovulation and the development of the Cl – even in cows. However, the physiological character of this passage (L359-366) is misleading and not suitable for a scientific review in bovines.

Author Response

Reviewer #2

  • The title reflects the content of the study. However, the title is no accurate and some chapters are unbalanced.
  1. We thank you; the title has been modified: Interval from oestrus to ovulation in dairy cows, a key factor for determining the insemination time: A review.

  • The OE-OY interval is a irrefutable physiological fact – only its consideration is a key factor for fertility within our ”artificial driven”  reproduction management. Figure 1 is a copy of a textbook diagram without critical reflection. A) The onset of oestrus does not start with the LH-peak. B)

AU: the fig.  has been modified.

  • The spermatozoa and oocyte lifespans are depended from many different factors; these parameter are not the core of this review

AU: We have changed the figure to make it clearer.  So, the information has been removed from the figure.

  • Therefore, the reviewer respect the very schematic character of this figure

AU: The figure 1 has been modified.

  • However, to rely on references which are more than 50-60 years old is questionable.

AU: We have reduced the number of references. About 20 “old” references have been reconsidered and most of them have been removed, such as Brewster,1941; Chenault, 1975; Laing, 1946; Hall, 1959; Studer, 1975 etc.

  • The authors raise the question regarding the conception rates following AI done twice or single a day without clear solution. A scientific review should improve real questionable practical situations. This chapter (L210-414) should be improved.

AU:  The chapter have been improved and references have been added.

  • Why should be done an US-examination of the ovulation every 3 hours (L244) ? It is interesting to evaluate the ovulation – however this interval is questionable as a key factor for fertility (title of this review). AU the sentence have been modify

AU: the sentence has been modified: “The ultrasound evaluation of ovulation time in practice may has some limitation because require a frequency of the examination that should occur at least twice a day”.

  • L 136 in combination with Fig. 1 is confusing.

AU: We have reorganized the figure to make it clearer.

  • The passages “sexed semen” (L220-225) and “inflammatory processes (L293-304) are unbalanced in comparison to the chapter “heat stress” and not identifiable in the subtitles. Should be extended or removed.
    AU: the passage “sexed semen” has been moved in chapter 4.1.

  • The application of hCG is an appropriate approach to stimulate the ovulation and the development of the Cl – even in cows. However, the physiological character of this passage (L359-366) is misleading and not suitable for a scientific review in bovines.

AU: the sentence lines have been removed.

Reviewer 3 Report

Comments and Suggestions for Authors

This is a potentially valuable contribution to a vital but well-worn topic. The authors need to clarify how this adds to the existing literature on the subject. For example, the authors could revise the narrative by emphasising recent contributions and critiquing these findings in greater detail.

Many of the sources cited are old. For example, line 71 is a key statement supported by references which are 27 and 31 years old. The currency of this information is therefore questionable.  

28 Change ‘during oestrus’ to ‘relative to oestrus’

34 Current wording implies that ovulation occurs during oestrus, which is inaccurate

84: Should read ‘… no differences’

136: This information does not align to figure 1, which shows the surge as starting within an hour of oestrus onset.

154: ‘Uterine tubes’ rather than ‘Fallopian tubes’

160-161: Unclear.

256: No evidence of the effect of photoperiod is presented. Provide support or omit this aspect.

293: Requires a new subheading

338-341: Clarify whether there is evidence to support a positive effect for the circumstances which are listed.

Comments on the Quality of English Language

English is generally of a good standard but lacks clarity in places. Language editing is required to improve clarity and ease of reading.

Author Response

Reviewer #3

  • This is a potentially valuable contribution to a vital but well-worn topic. The authors need to clarify how this adds to the existing literature on the subject. For example, the authors could revise the narrative by emphasising recent contributions and critiquing these findings in greater detail.

AU: Some very recent papers have been considered (Dalton et al., 2021, Koyama et al., 2023) and more comments have ben added throughout the paper.  Chapter 2.1 and 4.1 have been revised.

  • Many of the sources cited are old. For example, line 71 is a key statement supported by references which are 27 and 31 years old. The currency of this information is therefore questionable.  

AU: We have reduced the number of references. About 20 “old” references have been reconsidered and most of them have been removed, such as Brewster,1941; Chenault, 1975; Laing, 1946; Hall, 1959; Studer, 1975 etc.

  • 28 Change ‘during oestrus’ to ‘relative to oestrus’

AU: We have changed it.

  • 34 Current wording implies that ovulation occurs during oestrus, which is inaccurate

AU: We have removed the “stage of oestrus”.

  • 84: Should read ‘… no differences’

AU: Done.

  • 136: This information does not align to figure 1, which shows the surge as starting within an hour of oestrus onset.

AU: The figure has been modified.

  • 154: ‘Uterine tubes’ rather than ‘Fallopian tubes’

AU: We have changed the words.

  • 160-161: Unclear.

AU: The sentence has been re-drawn.

  • 256: No evidence of the effect of photoperiod is presented. Provide support or omit this aspect.

AU: We have chosen to omit the effect of photoperiod.

  • 293: Requires a new subheading:

AU: The information regarding the inflammatory process that can interfere with ovarian function and therefore, with the oestrus-ovulation interval has been moved in chapter 2.1.

  • 338-341: Clarify whether there is evidence to support a positive effect for the circumstances which are listed.

AU: The sentence has been removed. One sentence has been added to point 7,1:“GnRH treatment at oestrus significantly improved CR in cow under heat-stress conditions (Ullah et al. 1996, Rosemberg et al., 1991)[122, 129], in cows with a low BCS at AI and in primiparous cow (Kaim et al.., 2003)[61].”

English is generally of a good standard but lacks clarity in places. Language editing is required to improve clarity and ease of reading.

AU: We have correct English in all the text to improve clarity and ease of reading.

Round 2

Reviewer 2 Report

Comments and Suggestions for Authors

The authords improved the ms significantly. The reviewer has now only minor comments:

L 261: What means "differences in follicular growth" ? Do the authors reflect on morphological (size ? - normally not) or functional differences of preovulatory follicles? I would recommend to replace it by the more general term  distinctions/ variations in follicular development. 

L314-316 The reviewer agrees with content, however the style and the syntax of this phrase needs some rewriting.

Comments on the Quality of English Language

see above

Author Response

L 261: What means "differences in follicular growth" ? Do the authors reflect on morphological (size ? - normally not) or functional differences of preovulatory follicles? I would recommend to replace it by the more general term  distinctions/ variations in follicular development. We have replaced with variations in follicular development.

L314-316 The reviewer agrees with content, however the style and the syntax of this phrase needs some rewriting. L274-276 We have revised the phrase.